# Rodent Models of Lung Disease: A Road Map for Translational Research

**DOI:** 10.3390/ijms26178386

**Published:** 2025-08-28

**Authors:** Jerome Cantor

**Affiliations:** St John’s School of Pharmacy and Allied Health Sciences, St John’s University, 8000 Utopia Parkway Queens, Queens, NY 11439, USA; cantorj@stjohns.edu

**Keywords:** animal models, ALI, pulmonary emphysema, pulmonary fibrosis, hyaluronan

## Abstract

Animal models provide a controlled and reproducible environment for investigating the pathogenesis of human lung diseases. In many cases, the morphological changes associated with a particular model may resemble those seen in their human counterparts, but the corresponding biochemical events may differ, and their timeframe may be significantly reduced. Nevertheless, gaining insight into human disease mechanisms may be possible by employing experimental approaches that minimize the problems associated with extrapolating data from animal studies. Such strategies include using more than one model of a particular disease, employing different routes of administration of the injurious agent, using a variety of animal strains or species, or focusing on biochemical mechanisms common to both the animal model and its human counterpart. For example, rodent models that replicate elastic fiber injury in human pulmonary emphysema have been used to test aerosolized hyaluronan’s ability to slow the disease’s progression. The same models facilitated the identification of a new biomarker for pulmonary emphysema that may be a real-time indicator of therapeutic efficacy in clinical trials. Therefore, the appropriate use of these models can provide a necessary road map for designing appropriate dosages, delivery routes, timeframes, and endpoints in clinical trials of novel agents for the treatment of lung disease.

## 1. Introduction

Animal models are essential in investigating the genetic, environmental, and physiological factors contributing to lung disease. These models provide a controlled experimental environment for identifying significant mechanisms underlying lung injury and the subsequent repair processes [1,2]. One of the primary advantages of animal models is their ability to mimic the complex interactions of various influencing factors that lead to human lung diseases such as chronic obstructive pulmonary disease (COPD), interstitial lung fibrosis, and acute lung injury (ALI). The reproducibility of these animal models across diverse experimental conditions is critical in designing and implementing clinical studies involving novel therapeutic agents. They enable researchers to evaluate the pharmacokinetics and pharmacodynamics of drugs, ensuring that safety and effectiveness are established before human testing.

Rodents, especially mice and rats, are frequently used to study the mechanisms underlying various lung diseases, such as asthma, COPD, pulmonary fibrosis, and ALI [3,4,5,6]. Their relatively short lifespan and genetically manipulable systems allow researchers to investigate disease processes, including inflammation, fibrosis, and airway remodeling. Rodent models can be genetically modified to identify proteins associated with lung diseases or to develop specific phenotypes [7]. As shown in Table 1, the knockout of genes related to inflammation and cell proliferation, such as tumor necrosis factor alpha (TNF-α), interleukin-1β (IL-1β), and caveolin-1, may be employed to investigate their role in the progression of experimentally induced ALI. Other knockout models involving transforming growth factor-beta (TGF-β) and metalloproteinase 12 (MMP-12) can be used to study the pathogenesis of pulmonary fibrosis and smoke-induced emphysema, respectively, and deletion of surfactant genes provides a means of determining how these lipoproteins prevent lung injury related to infections.

While genetic models are commonly used to study specific mechanisms of lung disease, this review focuses on the role of chemically induced models. These models offer greater flexibility in manipulating a broad range of experimental conditions. They can be precisely controlled with regard to the concentration, duration, and timing of exposures to single or multiple toxins, providing consistent conditions to study disease mechanisms. This approach also eliminates the potential confounding influence of genetic manipulation, making it easier to establish causal relationships.

Regarding the use of particular species, rodent models are particularly valuable for studying the immune and inflammatory responses in the lungs [8]. They allow researchers to investigate the effects of various stimuli on lung function, including allergens and toxic chemicals. Techniques, such as imaging, biochemistry, and functional assays, can be employed in rodent models to evaluate lung function and disease progression in real time, providing more relevant data than in vitro studies [5]. Certain rodent strains can mimic specific human disease conditions more closely. For instance, using a mouse model of allergic asthma allows researchers to test responses to allergens and the efficacy of asthma therapies in a context that resembles human pathophysiology [9]. Furthermore, rodent models, rather than those involving larger mammals, may be more ethically acceptable while providing a practical advantage in terms of investigating potential therapies and mechanisms of lung injury over shorter timescales and at lower husbandry costs.

To further illustrate the critical role of rodent models in facilitating a deeper understanding of various lung diseases, this review focuses on how to transform the role of these models from their traditional role in describing morphological and biochemical changes to a more dynamic platform that can identify critical mechanisms of pulmonary injury, interactions of toxic agents, and potential treatment targets. Despite the potential disparities associated with extrapolating these types of studies to human disease, they provide a unique opportunity to develop novel strategies that permit early detection of lung disease and timely therapeutic intervention.

## 2. LPS Model of ALI

### 2.1. Biochemical and Morphological Features

ALI is an important feature of various pulmonary diseases, including pneumonia and respiratory distress syndrome [10,11]. Lipopolysaccharide (LPS), a component of Gram-negative bacteria’s outer membrane, potentiates the acute inflammatory response in both humans and animal models by inducing the release of proinflammatory cytokines that activate an inflammatory cascade that recruits additional immune cells to the lung (Figure 1) [12,13]. Several models of this type of injury involve either intratracheal, intraperitoneal, or intravenous administration of LPS to mice, rats, and hamsters. The intratracheal route is preferred because the acute inflammation is mainly limited to the lung, avoiding the confounding effects of systemic exposure to this agent.

LPS is recognized by the immune system primarily through pattern recognition receptors (PRRs), especially Toll-like receptor 4 (TLR4). Upon binding of LPS to TLR4, several intracellular signaling pathways are activated [14]. The binding of LPS to TLR4 leads to the activation of various immune cells, which release proinflammatory cytokines that play crucial roles in amplifying the inflammatory response, promoting further recruitment of immune cells to the site of injury [15].

The released cytokines also induce changes in the pulmonary endothelial cells, increasing permeability and contributing to edema formation [16]. This increased vascular permeability is a hallmark of ALI, leading to the accumulation of fluid in the alveolar spaces. This process is accompanied by an influx of neutrophils, which migrate to the site of injury in response to chemokines produced during the earlier phases of inflammation [17,18,19]. As lung injury progresses, several anti-inflammatory signals and mediators are produced, promoting the resolution of inflammation and repair of the lung tissue. However, in models involving multiple doses of LPS, the inflammatory response may remain unchecked, leading to alveolar wall injury and airspace enlargement [20].

The difference between single and multiple treatments with LPS has significant implications regarding extrapolation to human ALI. The transitory influx of neutrophils associated with a single treatment with LPS may not fully encompass the chronic lung injury mechanisms or repair processes relevant to the human condition [21,22]. This limitation is particularly applicable to lung histopathology, where repeated exposure to LPS can result in airspace enlargement and interstitial fibrosis. In contrast, a single exposure to this agent does not produce similar changes. Furthermore, not all ALI is induced by infections involving LPS, which may limit the model’s extrapolation to other forms of the disease.

Nevertheless, the LPS model provides a simple, reproducible means of determining the effects of anti-inflammatory and antifibrotic agents by allowing precise monitoring of drug effects on lung function parameters, histopathological changes, cytokine profiles, and immune cell dynamics [23]. This information is critical to developing dosing regimens and appropriate endpoints in clinical trials.

### 2.2. Synergistic Effect Between LPS and Cigarette Smoke

Prolonged exposure to cigarette smoke is known to contribute to chronic obstructive pulmonary disease (COPD), but its immediate effects remain less well understood [24]. Short-term inhalation of smoke does not typically result in significant lung damage and may only pose risks when there is pre-existing lung disease [25,26]. This is particularly relevant for second-hand smoke, whose impacts are often subtle and can trigger inflammatory responses only in the presence of concurrent lung injury.

The interaction between cigarette smoke and existing pulmonary conditions may involve activating common proinflammatory pathways [27,28,29]. For instance, the oxidants in cigarette smoke might enhance the effects of reactive oxygen species generated by secondary lung injuries [29]. Additionally, the increase in various cytokines resulting from smoke exposure could heighten the activity of a pre-existing population of inflammatory cells [30]. As a result, even brief exposure to second-hand smoke could exacerbate a latent disease process and make healthy lungs more susceptible to injuries from other agents.

To investigate these hypotheses, hamsters were exposed to second-hand cigarette smoke for two hours daily over three days, before or after administering lipopolysaccharide (LPS) intratracheally [31]. Findings indicated that short-term exposure to cigarette smoke heightened several inflammatory parameters in both scenarios. However, pretreatment with smoke had a greater effect than post-treatment, producing a 17% increase in inflammatory cell infiltrates and interstitial thickening (as measured on a scale of 0–4), a 25% rise in tumor necrosis factor receptor 1 (TNFR1)-labeled macrophages, a 27% increase in apoptotic cells, and a 284% elevation in bronchoalveolar lavage fluid (BALF) neutrophils.

The reasons for this discrepancy are unclear, but they may be related to the differing lung responses based on the timing of exposure to cigarette smoke and LPS. Given the multitude of toxins in cigarettes, pre-exposure might activate a wide array of inflammatory mechanisms, which are then further amplified by the endotoxin [32]. In contrast, pretreatment with LPS, a single toxin, may initiate a narrower range of inflammatory processes, thus reducing potential synergy with cigarette smoke.

The most striking difference between the pre- and post-smoked groups involved the influx of neutrophils in the lung. The marked increase in BALF neutrophils associated with pre-exposure to smoke is indicative of the increased migration of these cells from the pulmonary capillary network to the lung interstitium, which may result from enhanced secretion of neutrophil elastase induced by LPS [33]. It was previously shown that a deficiency in this enzyme significantly reduces the smoke-induced movement of neutrophils into the extravascular space, suggesting that elastase facilitates the dissociation of these cells from adhesion molecules that tether them to capillary endothelium (Figure 2) [34]. Studies have also shown that the development of pulmonary inflammatory infiltrates following either smoke exposure or LPS administration may be mainly due to the effects of TNF-alpha. The increased production of this cytokine by treatment with both agents may amplify the transcription of genes responsible for synthesizing matrix metalloproteinases and recruiting neutrophils to the lung [35,36].

While these studies concentrated on cigarette smoke, other inhaled pollutants could have similar effects. Both outdoor and indoor air contaminants may exacerbate underlying pulmonary diseases. Synergistic interactions involving even low levels of environmental toxins could sustain subacute inflammatory responses, potentially leading to significant lung damage over time (Figure 2).

### 2.3. Role of Endothelin in LPS-Induced Lung Injury

This laboratory has previously used the LPS model to determine whether the effects of LPS could be mitigated by decreasing the level of endothelin-1 (ET-1), a potent vasoconstrictor that is secreted by endothelial cells, vascular smooth muscle cells, and macrophages [37,38]. This mediator was chosen as a target molecule because it promotes the adhesion of neutrophils to endothelium and facilitates their migration into surrounding tissues (Figure 3) [39,40].

These studies showed that pretreatment with HJP272, a novel endothelin receptor antagonist (ERA), one hour before intratracheal instillation of LPS, significantly reduced a number of inflammatory parameters, including lung histopathological changes, TNFR1 expression by BALF macrophages, and alveolar septal cell apoptosis [41]. Conversely, exogenous administration of ET-1 increased LPS-induced lung injury, as measured by BALF neutrophils and TNFR1-positive macrophages [41].

The anti-inflammatory effects of an endothelin-converting enzyme inhibitor, phosphoramidon, were also studied in the LPS lung model using a similar group of inflammatory endpoints [42]. This agent prevents the conversion of a larger molecular weight precursor to ET-1. Pretreatment with this agent attenuated the effects of LPS to a greater extent than HJP272, which may be due to the fact that phosphoramidon completely blocks ET-1 activity, whereas HJP272 only binds to one of the two main endothelin receptors that regulate inflammatory cell influx into the lung [43,44].

In the case of both HJP272 and phosphoramidon, modifying the initial stages of the inflammatory process may be necessary to reduce the effects of LPS. Pretreatment with a single dose of HJP272 decreased a number of inflammatory processes, while administration of phosphoramidon at one hour following initiation of lung injury abrogated its inhibitory effect on the influx of neutrophils into the lung.

## 3. Elastase-Induced Pulmonary Emphysema

### 3.1. Biochemical and Morphological Features

In pulmonary emphysema, a subacute inflammatory process releases enzymes that slowly degrade alveolar wall elastic fibers, reducing their ability to store the energy needed to expel air from the lungs [45,46]. These fibers have a specialized structure consisting of a central elastin protein surrounded by layers of microfibrils [47]. The elastin component consists of cross-linked peptide chains containing hydrophobic domains that absorb energy by undergoing distention during inhalation and transitioning to a lower entropy state. The subsequent increase in entropy during exhalation releases the stored energy, producing the mechanical recoil that expels air from the lung [48]. Loss of these fibers due to enzymatic or oxidative breakdown causes uneven transmission of forces in the lung, leading to alveolar wall distension and rupture.

Rodent models of elastase-induced emphysema are widely used to study this mechanism of injury. There are a number of well-established models involving the use of a variety of elastases to degrade elastin and induce airspace enlargement in hamsters, rats, and mice [49]. The elastase can be administered via different routes, including intratracheal instillation or intrapulmonary delivery, which introduces the elastase to specific lung regions. Dosing protocols may involve a single dose or multiple doses of elastase, which prolongs lung injury and may involve additional inflammatory mechanisms [50,51]. Using a single intratracheal instillation of elastase generally results in a patchy distribution of airspace enlargement, whereas repeated treatments result in a more uniform pattern of injury (Figure 4). Nevertheless, the timeframe for developing airspace enlargement in these models is only a tiny fraction of that associated with the human disease. It cannot replicate the long-term consequences of interactions involving multiple inflammatory mechanisms that can vary according to environmental and genetic factors.

The model mimics human emphysema through the destruction of alveolar structures and an increase in airspace enlargement. Histological examination may reveal inflammatory cell infiltration, changes in lung architecture, and alterations in extracellular matrix components. Tissue samples may also be extracted for molecular analysis, including gene expression studies related to oxidative stress, inflammatory pathways, and proteolytic activities. Lung function tests may also assess airway resistance, compliance, and gas exchange efficiency.

Studies using these models often analyze BALF for inflammatory markers, cytokines, and cellular profiles. These biomarkers may be used to monitor lung injury and provide an early endpoint for therapeutic efficacy in clinical trials. One important biomarker that has been extensively investigated is desmosine and its isomer, isodesmosine (DID), which are elastin-specific crosslinks released from damaged elastic fibers [52]. Loss of these crosslinks causes unraveling and fragmentation of elastic fibers, leading to alveolar wall distension and rupture [53]. Consequently, elevated DID levels in body fluids may be an early indicator of alveolar wall distention and rupture.

### 3.2. The Ratio of Free to Total Desmosine as a Biomarker for Pulmonary Emphysema

Since various studies have shown that total DID levels may not be a sensitive biomarker for airspace enlargement, the ratio of peptide-free to peptide-bound DID was used, rather than the absolute quantity of these crosslinks, as an indicator of alveolar wall injury in a hamster model of elastase-induced pulmonary emphysema [54]. Peptide-bound DID refers to crosslinks attached to the elastin protein, whereas free DID is not linked to other amino acids.

The free-to-bound DID ratio was measured in BALF recovered from hamsters instilled intratracheally with either 5 or 10 U of elastase and controls given the saline vehicle alone [54]. Treating the lungs with different doses of elastase produced a range of emphysematous changes. It facilitated the comparison of the free-to-bound DID ratio with emphysematous changes, as measured by relative lung surface area. The results showed a statistically significant negative correlation between this ratio and lung surface area, which is inversely proportional to alveolar diameter (Figure 5). Clinical corroboration of this finding was seen in COPD patients, where the free-to-bound DID ratio showed a significant negative correlation with disease severity, as measured by forced expiratory volume [55]. In both cases, the relative increase in free DID may reflect the effects of both elastase activity and mechanical forces that facilitate the separation of intact crosslinks from their surrounding peptides.

### 3.3. Combining Elastase-Induced Emphysema with LPS

Acute exacerbations of chronic obstructive pulmonary disease (AECOPD) can lead to a rapid decline in lung function [56]. To investigate the impact of acute inflammation on existing elastic fiber damage, a hamster model was developed that utilizes intratracheal instillation of LPS to enhance pulmonary emphysema previously caused by elastase administration [57]. Unlike earlier models that involved either multiple elastase instillations followed by LPS treatment or a prolonged period between the administration of these agents, this model employs a single low dose of elastase with a reduced interval of just one week between the application of the two agents [58,59]. This design amplifies the effect of LPS relative to elastase and allows for the exploration of potential synergistic effects (Figure 6).

Using this model, it was possible to assess whether slight changes in the structure of elastic fibers made them more vulnerable to injury from LPS. Animals treated with elastase and LPS displayed prominent fragmentation and splaying of lung elastic fibers. Less pronounced changes were seen following treatment with either elastase or LPS alone. These findings were reflected by significantly higher levels of free BALF DID following treatment with both agents compared to the elastase-only group or untreated controls. Consistent with this process, a marked increase in airspace enlargement was seen in animals given both elastase and LPS compared to all other groups.

Additional studies examined the impact of elastin peptides obtained from human lung elastic fibers in an LPS model of ALI [57]. The simultaneous administration of elastin peptides and LPS led to marked increases in the percentage of BALF neutrophils as well as elevated levels of free BALF DID, in comparison to treatments with either substance alone. Following this, in vitro experiments were conducted using BALF macrophages from untreated hamsters to assess the chemotactic activity of elastin peptides [57]. When used separately, both elastin peptides and LPS boosted macrophage chemotaxis compared to control groups. However, the combination of these two agents resulted in a significantly greater chemotactic response than either treatment alone. These results support the concept of a self-reinforcing cycle of alveolar wall damage characterized by the recruitment of inflammatory cells induced by elastin peptides, increased degradation of elastic fibers, and the subsequent proliferation of these proinflammatory peptides (Figure 7).

## 4. Cigarette Smoke-Induced Pulmonary Emphysema

### 4.1. Biochemical and Morphological Features

Rodent models of cigarette smoke exposure are widely utilized in preclinical research to investigate the effects of tobacco smoke on health and to explore the underlying mechanisms of smoking-related diseases [60,61,62]. These models typically involve exposing rats or mice to either mainstream or sidestream cigarette smoke, simulating the inhalation patterns seen in human smokers. The exposure protocols can vary in duration, frequency, and smoke concentration to mimic different smoking behaviors, such as chronic smoking or occasional exposure.

Whole-body smoke exposure involves placing rodents in a chamber where they are subjected to smoke from lit cigarettes [63,64]. The chamber is designed to deliver controllable concentrations of smoke, ensuring uniform exposure. Parameters such as the number of cigarettes smoked per day, duration of exposure, and frequency can be adjusted to represent different smoking habits, from chronic daily smoking to intermittent exposure.

Local exposure is often employed to more accurately mimic the inhalation process. It can involve direct exposure to smoke through a tube or other apparatus, allowing researchers to control the inhalation pattern more precisely, particularly with regard to quantifying total smoke particulates. Alternatively, cigarette smoke condensate or extract may be directly delivered to the lungs via intratracheal injection [65]. This method allows researchers to investigate the effects of specific components of cigarette smoke without the variability introduced by inhalation.

Certain strains of mice are commonly used because of their increased susceptibility to smoke-induced oxidant injury [66]. Exposed animals exhibit physiological changes similar to those seen in human smokers, including increased sensitivity of the airways to various stimuli [60]. Both humans and mice show bronchial wall changes consisting of inflammation, epithelial metaplasia, and proliferation of extracellular matrix [67,68]. Distal lung alterations include airspace enlargement occurring over a period of several months and accumulation of smoke particles in alveolar macrophages. The gradual progression of these changes more closely resembles the human pulmonary emphysema than the acute airspace injury associated with the elastase model of the disease.

Mice treated with whole-body exposure to cigarette smoke for 2 consecutive hours each day, 5 days per week, over 10 months showed airspace enlargement that became microscopically evident at 2 months [69]. Total lung DID was increased at two months, then markedly decreased over the next four months before undergoing a secondary increase over the remaining course of the study. These findings reflect a dynamic balance between elastic fiber injury and repair over time and are consistent with the progressive decline in BALF DID levels from 4 to 10 months.

Airspace enlargement leveled off in the smoke-exposed mice after 4 months, possibly related to increased lung DID content. Enhanced deposition of elastin and other extracellular matrix components could decrease alveolar wall rupture due to mechanical stress. This hypothesis is supported by clinical studies showing an association between interstitial pulmonary fibrosis and cigarette smoking [70].

### 4.2. The Effect of Aerosolized Hyaluronan on Airspace Enlargement

Although the investigation of potential agents for treating pulmonary emphysema has focused on elastase inhibitors, this laboratory has developed a novel method of preventing elastic fiber injury by administering aerosolized hyaluronan (HA). Hamsters exposed to HA before intratracheal instillation or elastase had significantly less airspace enlargement than untreated controls [71]. This effect may be due to the attachment of HA to lung elastic fibers, which provides a protective coating against elastases and other injurious agents [72].

Studies using the same model showed that pretreatment with intratracheally administered hyaluronidase significantly increased airspace enlargement, providing the rationale for investigating the potential therapeutic effects of HA [71]. Initially, the HA was administered intratracheally immediately after elastase instillation, but an aerosolized preparation of this agent has a similar effect when administered for 50 min prior to elastase treatment.

To further test this concept, the effect of aerosolized HA was studied in a mouse model of cigarette smoke-induced pulmonary emphysema. The smoking model provides a more stringent test of the therapeutic effects of HA because airspace enlargement develops over a period of months and more closely resembles the development of the human disease [73]. Mice were treated with an aerosolized solution of 0.1 percent HA (average molecular weight of 150 kDa) before a 2-h exposure to cigarette smoke. The procedure was repeated 5 days per week over 6 months. While smoke-induced airspace enlargement leveled off over time, animals treated with HA nevertheless showed a significant reduction in airspace enlargement during the study (Figure 8).

To determine the location of the HA within the lung, animals were treated with a single 1-h exposure to fluorescein-labeled 150 kDa HA. Following treatment, a linear fluorescence pattern was associated with interstitial, pleural, and vascular elastic fibers (Figure 9). At 24 h, the fluorescence was diminished in intensity, consistent with the clearance of the exogenous HA from the lung [72,73]. These findings supported earlier unpublished studies showing that the aerosol droplet size associated with low molecular weight HA more readily penetrates the distal lung airspaces.

These findings provided the rationale for translational studies involving a clinical trial to determine the efficacy of HA in COPD patients with alpha-1 antiprotease deficiency [74]. Levels of DID in plasma, urine, and sputum were measured at weekly intervals to assess the effect of treatment on lung elastic fiber degradation. Inhalation of a 0.01 percent solution of aerosolized HA twice daily for 28 days significantly decreased the amount of free DID in urine. This finding was consistent with earlier measurements of free BALF DID in elastase-induced pulmonary emphysema, supporting its use as a biomarker for therapeutic efficacy. Such real-time indicators of drug effects can provide an early decision point for the potential termination of clinical trials that might otherwise be prolonged due to their reliance on less sensitive markers such as pulmonary function tests or questionnaires assessing patient well-being.

## 5. Bleomycin Model of Pulmonary Fibrosis

### 5.1. Biochemical and Morphological Features

The rodent bleomycin (BLM) model is a widely used experimental model to study lung fibrosis and pulmonary diseases, particularly idiopathic pulmonary fibrosis. In this model, bleomycin, a chemotherapeutic agent and antibiotic, is administered to rodents, typically mice or rats, to induce lung injury and fibrosis [75,76]. The agent can be delivered via various routes, such as intranasally and intratracheally. A single dose or a series of doses over several days is given. The concentrations and timing may vary depending on the desired severity of fibrosis, with high doses causing more acute injury and potentially more pronounced fibrotic changes.

The bleomycin (BLM) model has morphological features that resemble the human disease, including an influx of inflammatory cells, alveolar epithelial cell hyperplasia, airspace distention, and interstitial fibrosis [77]. Intratracheal instillation of BLM induces the formation of complexes between this agent and Fe^2+^, which generate free radicals that damage DNA, leading to necrosis of type 1 alveolar epithelium and exposure to underlying basement membranes (Figure 10) [78]. This process is accompanied by a marked influx of neutrophils, proliferation of type 2 alveolar cells, and the deposition of collagen and other extracellular components in the lung interstitium [75].

The rapid development of inflammation and fibrosis following the instillation of BLM suggests that this form of lung injury may be better characterized as a wound-healing phenomenon rather than irreversible lung remodeling. The changes induced by BLM do not replicate the gradual development of the human disease and can regress over time (Figure 10) [79]. Nevertheless, this model has provided insight into the mechanisms that may be responsible for the development of its human counterpart. The rapid development of fibrosis facilitates the evaluation of drug candidates over the entire course of the disease. By varying the temporal relationship of these agents to BLM instillation, it may be possible to determine where they exert their most significant effect on the disease process.

While rodent models provide useful insights, there are inherent differences between rodent and human biology that can affect the translation of findings to human disease. The effects of bleomycin can vary widely based on dosing, route, and the specific strain of rodents used, requiring careful standardization of experimental protocols to reduce the confounding effects of these variables.

### 5.2. The Effect of Endothelin Inhibition

To evaluate the temporal relationship between BLM-induced pulmonary fibrosis and treatment with HJP272, hamsters were given an intraperitoneal injection of this agent one hour before intratracheal instillation of BLM or 24 h afterward [80]. During the following month, the pulmonary inflammatory response was assessed by measuring various parameters, including lung histopathological changes, BALF neutrophil content, lung collagen content, BALF macrophage TNFR1, and alveolar septal cell apoptosis.

Pretreatment with HJP272 resulted in a significant decrease in all these parameters compared to animals receiving BLM alone, whereas post-treatment was ineffective in reducing their levels. This discrepancy was most evident morphologically, where the lungs of animals pretreated with HJP272 showed much less fibrosis, suggesting that the early inflammatory events may be primarily responsible for the extent of lung injury. These findings are consistent with clinical trials showing that commercially available ERAs, such as ambrisentan and bosentan, are ineffective in treating pre-existing pulmonary fibrosis [81,82,83]. However, this finding does not preclude the use of ERAs as prophylactic agents, which may be given in combination with drugs whose side effects include pulmonary fibrosis.

The limited effectiveness of ERAs may be related to the emergent characteristics of pulmonary fibrosis, where interactions across various scales lead to a spontaneous reorganization of lung structure [84,85]. Emergence is a hallmark of complex systems, such as chemical reactions, epidemics, and disease progression. This phenomenon can be illustrated by percolation models that describe the random movement of fluids through interconnected channels. As isolated currents within this network converge and reach a critical threshold, a phase transition occurs, altering the system’s structure and behavior [86].

In pulmonary fibrosis, the expansion of the extracellular matrix within the lung interstitium resembles the diffusion of fluid through a percolation network, resulting in similar shifts in the lung’s chemical and physical properties [87]. The accumulation of matrix components changes the elastic modulus of the alveolar walls, affecting how mechanical forces related to breathing are transmitted. This alteration leads to additional interstitial damage and repair, fostering the disease’s self-perpetuation on a larger scale (Table 2). The resulting morphological changes include interstitial fibrosis with cystic dilation of airspaces.

Computer-generated models of pulmonary fibrosis support this mechanism, demonstrating how localized changes in the alveolar wall structure can give rise to widespread morphological transformations that mirror those found in the human condition [87]. The concept of emergence underscores the importance of developing biomarkers that possess the sensitivity and specificity to identify and address the disease before it leads to more extensive lung damage that could benefit from treatment. Identifying such a biomarker could facilitate the timely administration of ERAs, potentially improving their therapeutic effectiveness.

## 6. Analyzing the Microenvironment of Rodent Lung Disease Models

While animal models enable the study of specific disease mechanisms, much of the information they provide involves larger-scale relationships between inflammatory mediators, constitutive protein synthesis, and morphological changes. Most models are less useful at lower levels of scale, particularly with regard to identifying genetic alterations at the cellular level by analyzing RNA.

Previously, bulk RNA sequencing was employed to average the gene expression information across many cells, but this approach can obscure important biological variations and heterogeneity within a cell population. The development of single-cell RNA sequencing (scRNA-seq) circumvents this limitation by providing a means of identifying the patterns of genetic expression within diverse cell populations, such as epithelium, fibroblasts, and immune cells [88,89,90]. This technique has significantly increased our understanding of the cellular microenvironment during disease progression. For example, in a mouse model of asthma, scRNA-seq can reveal the expansion of specific immune cell subsets, such as eosinophils or T helper cells, and how their activity correlates with disease severity. Comparing the scRNA-seq data from healthy and diseased animals can also identify cell types or states associated with a particular lung disorder. In models of pulmonary fibrosis, scRNA-seq can uncover the activation of fibroblasts and their transition into pathological myofibroblasts [88]. This information can facilitate the development of novel therapeutic approaches to preventing or reversing fibrosis.

Furthermore, scRNA-seq can be used to evaluate the efficacy of therapeutic interventions [91]. By profiling cells before and after treatment, researchers can assess how therapies modulate the expression of genes associated with inflammation, repair, or immune responses. In evaluating new drugs for asthma, comparing the scRNA-seq profiles of treated versus untreated animals can identify the mechanisms through which the drug exerts its effects, providing a pathway for translating findings from animal models to human disease [92].

## 7. Animal Model Alternatives

While animal models have traditionally been a fundamental component of biomedical research, the ethical concerns and high costs associated with animal studies have driven the search for alternative methods that can provide reliable and relevant data (Table 3). None of these approaches alone can provide an adequate substitute for animal models, but when combined with other procedures, they can produce complementary findings that reveal patterns of injury that may be extrapolated to human disease.

The use of various cell culture techniques provides a means of investigating specific mechanisms of lung injury in greater detail. Primary human bronchial epithelial cells and immortalized cell lines have been utilized to study individual cell responses to various stimuli, including allergens, pollutants, and pharmaceuticals [93,94,95]. Furthermore, the development of three-dimensional cell culture systems provides a more physiologically relevant model compared to traditional two-dimensional cultures [96,97]. They may be used for high-throughput screening of drug efficacy and toxicity while providing real-time monitoring of cellular responses. In particular, patient-derived organoids and lung-on-a-chip devices play an important role in studying patient-specific lung diseases [94].

A hybrid approach that bridges the gap between animal models and tissue culture involves the use of ex vivo techniques, such as isolated, perfused lungs and precision-cut lung slices. In both cases, the architectural features of intact lungs are preserved, including the spatial relationship of individual cells and to their surrounding extracellular matrix. Consequently, the findings derived from these techniques may be more readily translated to the clinical setting than those obtained by the use of in vitro procedures alone.

In addition to tissue-based systems, advancements in computational biology and bioinformatics are enabling the development of robust in silico models to simulate lung diseases. Regarding the pathogenesis of pulmonary emphysema, in silico studies have shown that localized variations in alveolar wall elasticity can coalesce into widespread morphological changes mimicking those observed in pulmonary emphysema [98,99]. These findings demonstrated that pulmonary emphysema may be an emergent phenomenon, in which complex interactions across multiple levels of scale lead to spontaneous reorganization of pulmonary architecture. This concept has the potential to improve the detection and treatment of this disease.

## 8. Strategies for Addressing the Limitations of Animal Models

Based on the examples presented in the current review, animal models play a critical role in investigating lung disease. Nevertheless, models involving a single dose of a toxic agent may not accurately represent the corresponding human disease [2]. The compression of the timeline of the morphological changes in these models cannot reproduce the long-term evolution of changes at multiple levels of scale that are an essential characteristic of pulmonary emphysema, interstitial fibrosis, and other lung disorders [100,101,102].

In the case of the BLM model of pulmonary fibrosis, the regression of disease after several months may confound the interpretation of the effects of therapeutic agents. This issue was demonstrated with the use of ERAs to mitigate BLM-induced lung injury. While these agents were effective in this model, the results could not be repeated in clinical trials [81,82,83]. Furthermore, anti-elastases that reduce experimentally induced pulmonary emphysema have had only limited success in the clinical setting, perhaps due to the interconnection of multiple human lung disease mechanisms that may circumvent the inhibition of a single pathway [103,104].

The selection of the rodent species is another critical factor in evaluating potential therapeutic interventions. In the case of the LPS-induced model of ALI, the use of certain inbred species of mice and rats to produce more uniform results can also increase the risk of pneumonia and other conditions, which can rapidly affect other animals in the same cage [105]. Conversely, the use of outbred strains, such as Syrian hamsters or Sprague–Dawley rats, can avoid this potential confounding factor. The choice of species is also important for the successful replication of human disease. For example, hamsters may be more susceptible to the damaging effects of elastase than rats, which have higher levels of alpha-1 antiprotease [106]. Furthermore, the use of mouse strains with lower levels of certain antioxidants can increase the progression of smoke-induced lung injury [107].

Despite the potential limitations imposed by the use of certain toxic agents and rodent species, the strength of animal models involves the ability to study mechanisms of injury from different standpoints (Table 4). Instead of focusing on descriptive studies that may involve epiphenomenal findings rather than pathogenetic processes, these models may be subjected to interventional treatments that identify the temporal relationship between specific molecular changes and the progression of lung disease. In this regard, the LPS model has been used to define the temporal relationship of neutrophil recruitment to subsequent lung injury.

Similarly, the administration of therapeutic agents may be temporally modified to determine whether their efficacy is dependent on administration at the beginning of lung injury or may be delayed until the disease is more advanced. This approach was used to determine whether aerosolized HA remains effective in limiting airspace enlargement when delayed for a month after initiation of cigarette smoke exposure [69]. The results of that study showed that the protective effect of HA is not dependent on immediate treatment and may therefore be appropriate for use in humans already diagnosed with pulmonary emphysema.

The use of multiple models to determine the consistency of responses to therapeutic agents is an example of this process. The findings associated with the use of HJP272 in the BLM model were replicated in another model of pulmonary fibrosis induced by the cardiac antiarrhythmic agent, amiodarone [108]. Since this model involves different mechanisms of injury than the BLM model, the similarity in results is more meaningful regarding the therapeutic potential of ERAs to mitigate human pulmonary fibrosis.

Furthermore, varying the routes of administration of the disease-inducing agent can yield important insights into the systemic factors involved in the development of human lung disease. Intratracheal versus intravenous instillation can lead to disparate responses in terms of the lung cells affected and the timeline of injury and repair. While intratracheal instillation may be more effective in producing lung changes, other toxin or drug delivery routes may better reflect the clinical circumstances surrounding the development and potential treatment of the human disease. In particular, the effect of endothelial cell injury on the influx of inflammatory cells and protein-rich fluid into the lung may be better studied using a model of lung injury induced by intravenous or intraperitoneal administration of toxic agents [4].

The current review shows that animal models are especially suited to investigating the potential interactions of pulmonary toxins. Pretreatment with brief exposure to cigarette smoke significantly enhanced LPS-induced lung inflammation [31]. Further investigation of this phenomenon revealed that exogenously administered endothelin facilitated the influx of neutrophils that may be sequestered in pulmonary capillaries following smoke exposure [109]. This finding suggests that LPS-induced activation of endothelin may be responsible for the synergy between this agent and cigarette smoke.

Pretreatment with a low dose of elastase also had a synergistic effect on LPS-induced lung inflammation [57]. This finding prompted the use of elastin peptides to determine their proinflammatory activity in the LPS model. Combining the two agents significantly increased BALF levels of neutrophils and free DID compared to either agent alone. These results emphasize the proinflammatory role of elastin peptides released from degraded elastic fibers and their ability to potentiate the effects of other injurious agents, such as pathogenic organisms and environmental pollutants. The synergistic interactions revealed by the combined use of elastin peptides and LPS may provide a better understanding of how acute exacerbations in patients with COPD may lead to the permanent loss of lung function.

## 9. Conclusions

Animal models have emerged as an invaluable tool in understanding the complex pathogenesis of various human lung diseases. These models serve as critical platforms for exploring the multifactorial mechanisms underlying these disorders. By illustrating various experimental approaches in multiple models of pulmonary disease, the current review emphasizes the role of animal models in identifying potential synergic interactions between toxic agents and how the temporal dynamics of exposure can critically influence the severity of lung injury. The continued refinement of these models will improve their relationship to human pulmonary diseases and facilitate the translation of experimental findings to therapeutic interventions that can slow the progression of lung injury and reduce the risk of respiratory failure.

## Figures and Tables

**Figure 1 ijms-26-08386-f001:**
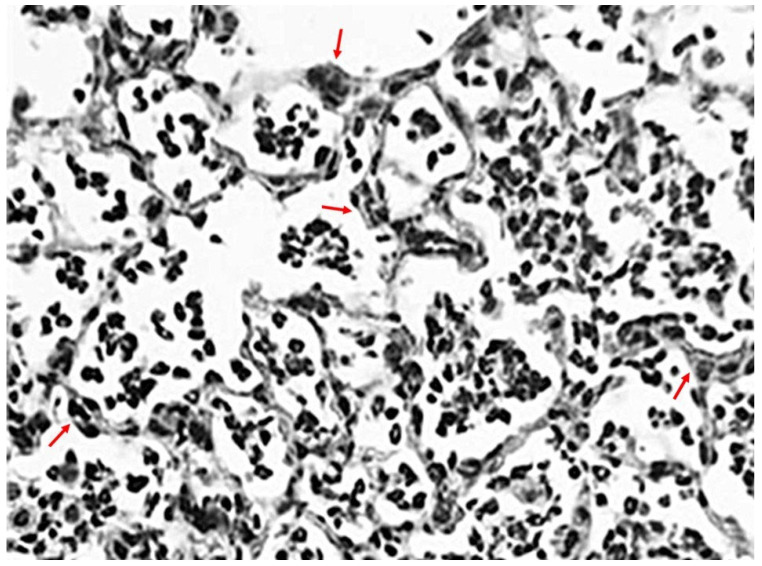
Photomicrograph of LPS-treated hamster lung at 24 h, showing a marked influx of neutrophils into alveolar spaces and focal interstitial thickening (arrows) following a single intratracheal instillation of this agent. The population of these cells peaks between 24 and 48 h, then rapidly declines. However, the presence of these cells can be maintained with multiple instillations of LPS, thereby mimicking human ALI more closely—Hematoxylin and Eosin; Original magnification: 200×.

**Figure 2 ijms-26-08386-f002:**
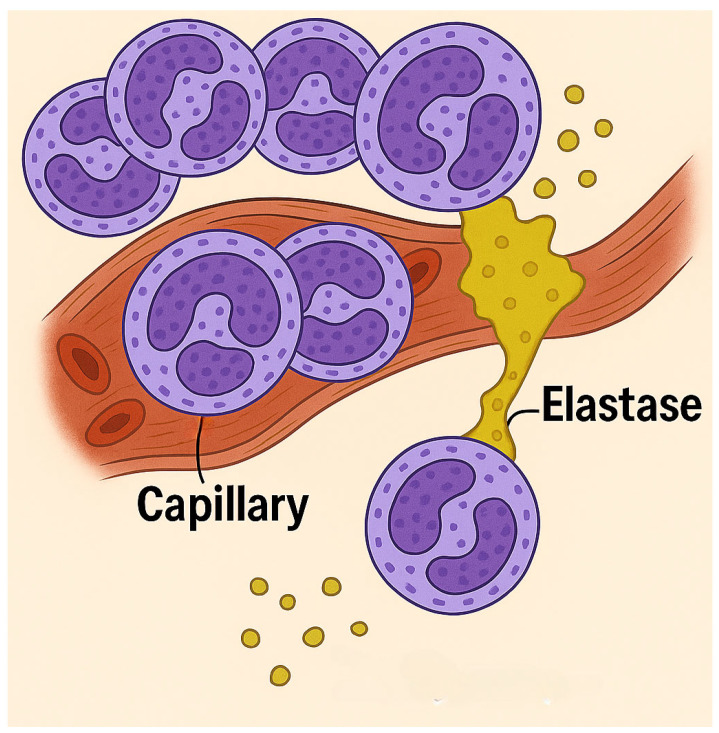
Illustration showing the potential mechanism responsible for the synergy between cigarette smoke and LPS. Following smoke exposure, neutrophils sequestered in alveolar capillaries are activated by LPS, inducing the secretion of elastases that increase their migration into the lung. A deficiency of elastase was previously shown to significantly decrease smoke-induced movement of neutrophils into the extravascular space, supporting the concept that elastase separates these cells from capillary endothelium.

**Figure 3 ijms-26-08386-f003:**
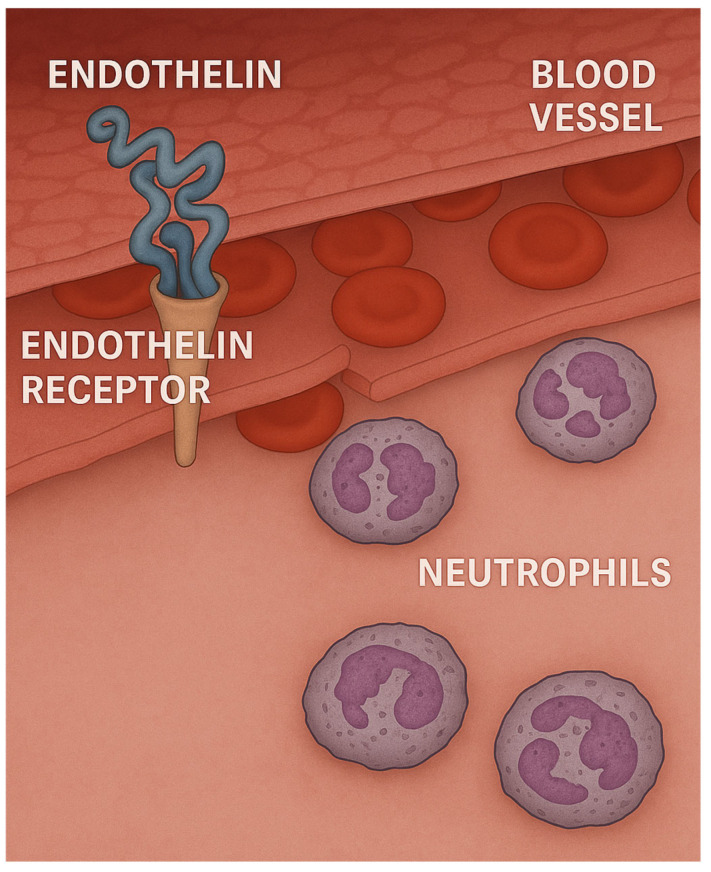
Endothelin may facilitate the influx of neutrophils into the lung. This concept is supported by studies showing that lung injury induced by intratracheal instillation of LPS is mitigated by pretreatment with an ERA (HJP272).

**Figure 4 ijms-26-08386-f004:**
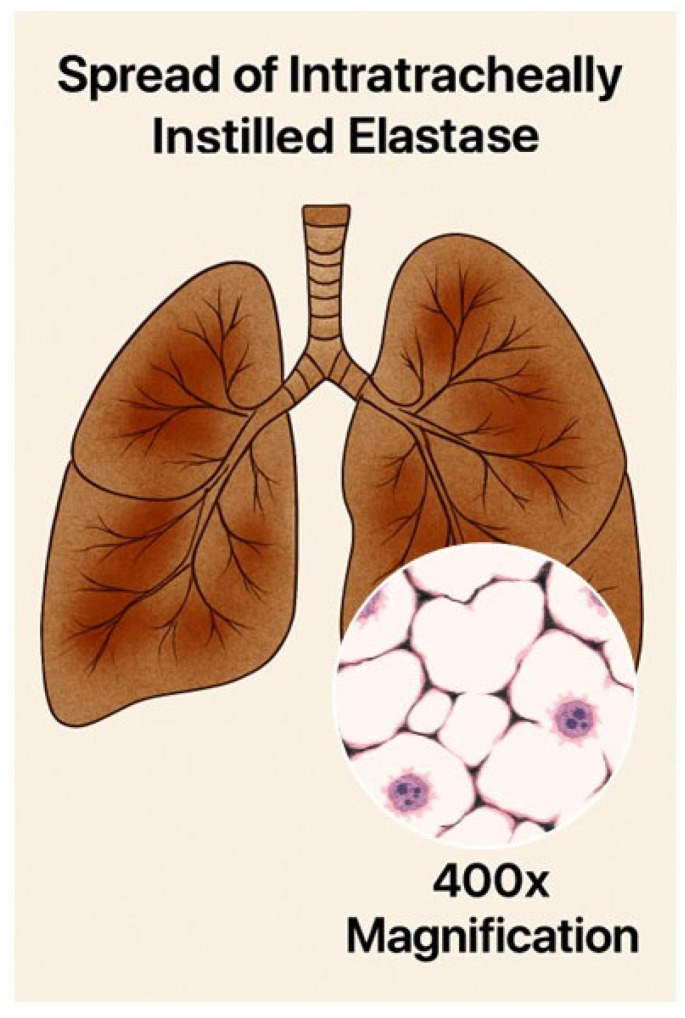
Illustration showing the patchy distribution of elastic fiber injury and airspace enlargement associated with a single intratracheal elastase instillation. The insert depicts a magnified area of lung tissue with fragmented elastic fibers (black lines) and alveolar wall rupture.

**Figure 5 ijms-26-08386-f005:**
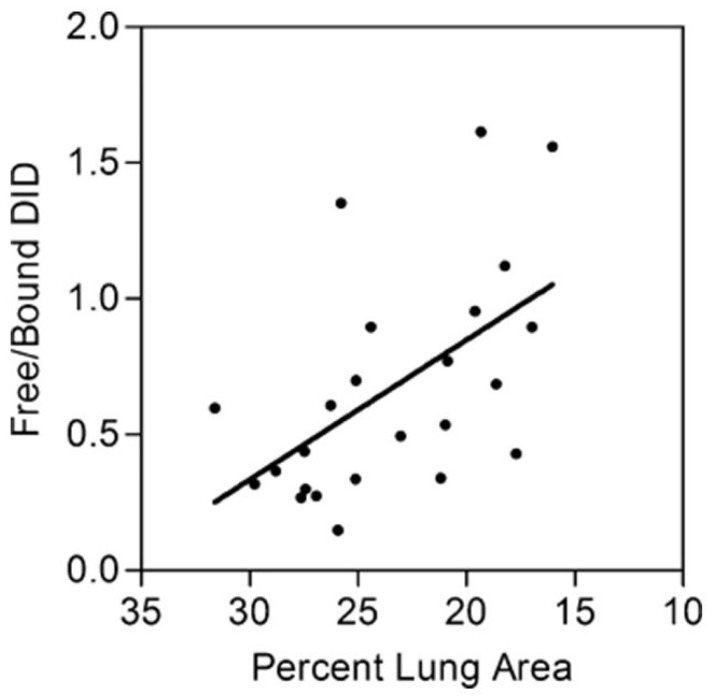
Graph showing the correlation between elastase-induced loss of lung surface area and the ratio of free to bound BALF DID following intratracheal instillation of either 5 or 10 units of porcine pancreatic elastase (r = −0.56; *p* < 0.01). It is hypothesized that the combination of elastase activity and mechanical strain causes the release of intact DID crosslinks from elastin peptides. This process increases the free-to-peptide-bound BALF DID ratio in hamsters intratracheally instilled with elastase [54].

**Figure 6 ijms-26-08386-f006:**
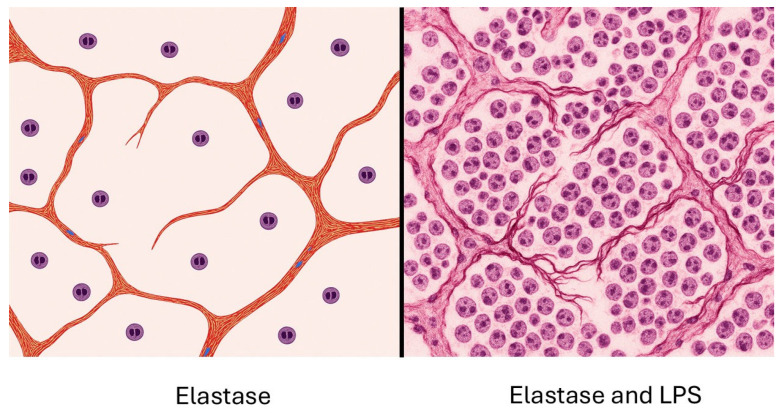
Illustration showing mild alveolar wall injury and rupture following intratracheal instillation of low-dose elastase (**left**). The instillation of LPS one week later results in an influx of neutrophils, which produce elastases and oxidants that increase elastic fiber damage, alveolar wall rupture, and airspace enlargement (**right**).

**Figure 7 ijms-26-08386-f007:**
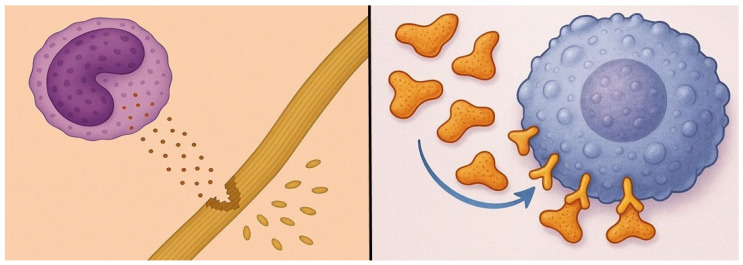
Elastin peptides released from damaged elastic fibers (**left**) bind to elastin receptor complexes on macrophages (**right**), inducing the release of proinflammatory cytokines. This mechanism results in a self-perpetuating inflammatory reaction.

**Figure 8 ijms-26-08386-f008:**
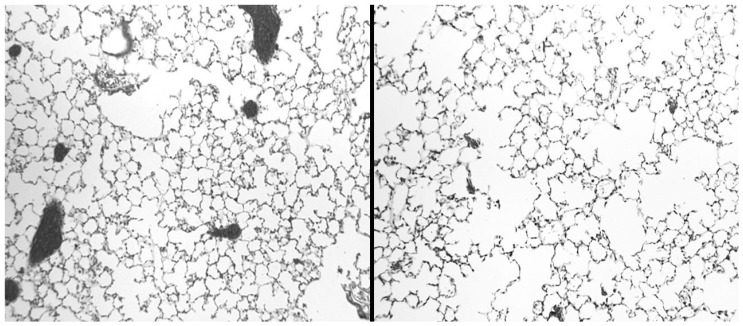
Photomicrographs of mouse lungs after 3 months of cigarette smoke exposure. Animals treated with aerosolized HA (**left**) had significantly less airspace enlargement than smoke-exposed animals given aerosolized saline (**right**). Hematoxylin and Eosin; original magnification: 100×.

**Figure 9 ijms-26-08386-f009:**
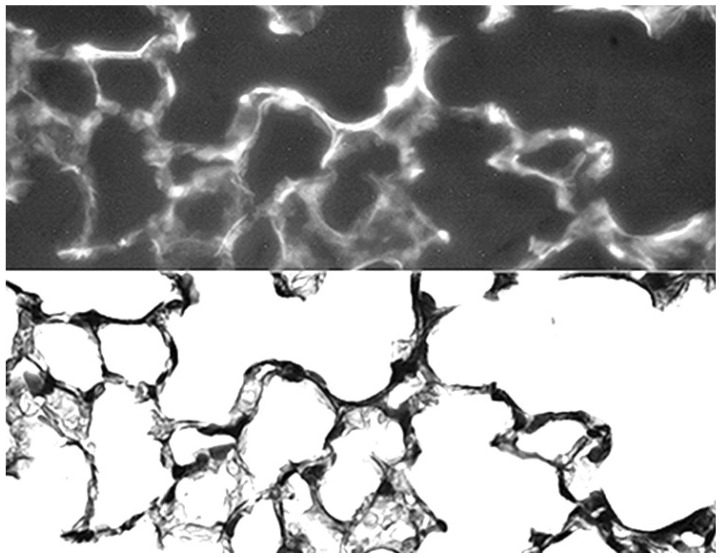
Photomicrographs of mouse lung at 24 h following intratracheal instillation of fluorescein-labeled HA. There was prominent fluorescence associated with elastic fibers (**upper**), which was confirmed by a photomicrograph of the same area stained for elastic fibers (**lower**). Reprinted with permission [73]. Original magnification: 400×.

**Figure 10 ijms-26-08386-f010:**
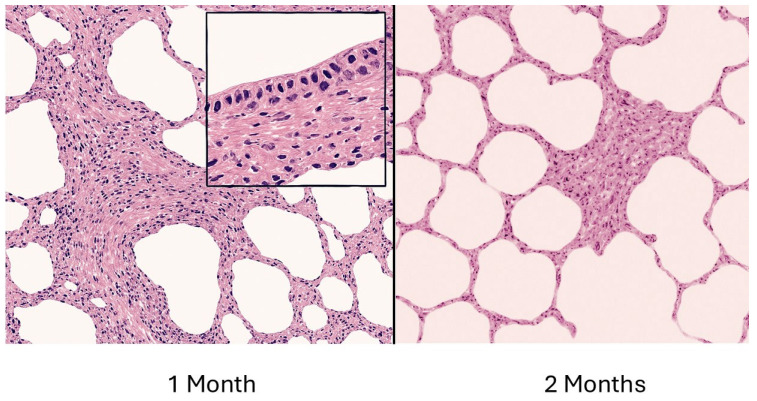
Illustration showing the evolution of BLM-induced pulmonary fibrosis. The fibrotic reaction is maximal around 1 month then gradually recedes. Morphological changes include interstitial fibrosis, cystic airspace dilation, and bronchial epithelialization of distal airspaces (insert).

**Table 1 ijms-26-08386-t001:** Rodent Knockout Models of Lung Disease.

Knockout Gene	Function	Phenotypic Outcomes	Disease Relevance	Common Applications	References
TGF-β	Fibrogenic signaling	Enhanced fibrosis; alveolar thickening	Pulmonary fibrosis	Studying ECM deposition and fibroblast activation	Calthorpe et al. Am J Physiol Lung Cell Mol Physiol. 2023;324(3):L285–L296. doi: 10.1152/ajplung.00106.2021
Elastase	Proteolytic tissue degradation	Alveolar destruction; emphysematous changes	Emphysema	Role of elastase in pathogenesis of emphysema	Suzuki et al. Am J Physiol Lung Cell Mol Physiol. 2020;318(6):L1172–L1182 doi: 10.1152/ajplung.00214.2019
IL-1	Pro-inflammatory cytokine	Reduced inflammation and immune cell infiltration	COPD, acute inflammation	Analyzing cytokine cascade and immune modulation	Malireddi et al. Front Immunol. 2022; 13:1068230 doi: 10.3389/fimmu.2022.1068230
MMP-12	Extracellular matrix remodeling	Impaired elastin breakdown; altered alveolar repair	Emphysema	ECM turnover and elastase activity studies	Lagente et al. Expert Opin Ther Targets. 2009:287–95 doi: 10.1517/14728220902751632
SP-A/SP-D	Surfactant proteins, innate immunity	Impaired host defense; surfactant dysfunction	Infection, alveolar injury	Surfactant biology and microbial clearance	Watson et al. Front Immunol. 2021 Jan 19;11:622598 doi: 10.3389/fimmu.2020.622598
Caveolin-1	Endothelial signaling and caveolae	Vascular leakage; exaggerated fibrosis response	Fibrosis, vascular remodeling	Vascular permeability and fibrosis progression	Fan et al. Front Pharmacol. 2024;15:1417834 doi: 10.3389/fphar.2024.1417834
TNF-α	Cytokine mediator of inflammation	Blunted inflammatory response; altered apoptosis	Chronic inflammation	Drug response and immunomodulation studies	Redente et al. Am J Respir Cell Mol Biol. 2014;50(4):825–37 doi: 10.1165/rcmb.2013-0386OC

**Table 2 ijms-26-08386-t002:** Emergent Properties of Pulmonary Fibrosis with Cystic Airspaces.

Stage	Scale	Key Mechanisms	Emergent Phenomena	Outcome in Lung Architecture
Initial Strain	Biomechanical	- Localized fiber overextension - Shear stress on fiber networks	Localized fiber fraying and interstitial rupture	Subtle weakening of septal integrity
Cellular Response	Cellular	- Epithelial-mesenchymal transition (EMT) - Fibroblast recruitment	Fibroblast recruitment and matrix remodeling	Early stiffening and remodeling of tissue zones
Molecular Escalation	Molecular	- TGF-β signaling - ROS accumulation - Piezo channel activation	Signal amplification and sustained matrix production	Loss of compliance and architecture
Feedback Intensification	Multiscale	- Altered mechanical load distribution - Time-dependent remodeling	Cyclic progression of strain and remodeling	Fibrotic stiffening begins maladaptive self-reinforcement
Structural Collapse	Biomechanical	- Elastic fiber tearing and fragmentation - Airspace dilation	Architectural failure and airspace dilation	Fibrotic consolidation and emphysematous expansion
Systemic Emergence	Systems-Level	- Cross-talk between immune, mechanical, and signaling systems	Disease-wide progression, system-wide “order”	Irregular lung patterning and irreversible damage

**Table 3 ijms-26-08386-t003:** Rodent Models of Lung Injury.

Category	Model	Advantages	Limitations	Rodent Species Other Sources	References
In Vivo –Chemical Induction	Bleomycin-induced fibrosis	Mimics human fibrosis; widely studied	Reduction in fibrosis over time	C57BL/6 mice, Sprague- Dawley rats; Syrian hamsters	Ishida et al. 2023 [77] Int J Mol Sci. 2023;24(4):3149 doi: 10.3390/ijms24043149
	Elastase-induced emphysema	Direct alveolar wall injury; dose-related	Limited inflammation; fast onset	C57BL/6 mice, Wistar rats; Syrian hamsters	Joshi et al. 2023 [50] Sci Rep 13, 2023; 15259 doi: 10.1038/s41598-023-41527-1
	LPS-induced acute lung injury	Strong immune response; acute model for ARDS	Limited chronic mimicry	BALB/c, C57BL/6 mice, Wistar rats; Syrian hamsters	Domscheit et al. 2020 [4] Front Physiol. 2020;11:36 doi: 10.3389/fphys.2020.00036
	Cigarette smoke exposure	Closely mimics human emphysema; includes oxidative stress	Long exposure periods; delivery variability	C57BL/6 mice, BALB/c mice, Wistar rats	Yao et al. 2008 [66] Am J Physiol Lung Cell Mol Physiol. 2008;294(6):L1174-L1186. doi: 10.1152/ajplung.00439.2007
In Vitro–2D Cultures	Lung cell stretch	Reproducible; controlled mechanical insult	Limited cell diversity; lacks feedback	Primary cultures and cell lines from various species	Martin-Vicente et al. Am J Respir Cell Mol Biol.; 72(2):195–205 doi: 10.1165/rcmb.2023-0449OC
	Air–liquid interface cultures	Preserves polarity; suitable for aerosol studies	Culture-sensitive; short lifespan	Primary cultures and cell lines from various species	Cao et al. 2021 [93] In Vitro Cell Dev Biol Anim. 2021;57:104-132. doi: 10.1007/s11626-020-00517-7
In Vitro–3D Systems	Lung organoids	Multicellular dynamics; native-like structure	Variable differentiation; perfusion limits	Humans and other species	Kühl et al. 2023 [94] Cells. 2023;12(16):2067. doi: 10.3390/cells12162067
	Lung-on-a-chip	Human relevance; mechanical/fluid integration	Technical complexity; limited throughput	Human cells	Francis et al. Drug Discov Today. 2022:9:2593–2602. doi: 10.1016/j.drudis.2022.06.004
Ex Vivo Systems	Isolated perfused lung	Preserves whole-organ architecture; real-time monitoring	Viability is short; setup intensive	Rats, mice, and other species	Eriksson et al. AAPS J. 2020;22(3):71 doi: 10.1208/s12248-020-0456-x
	Precision-cut lung slices	Spatial structure retained; ECM studies	Static system; fresh tissue required	Rats, mice, and other species	Lam et al. Front Pharmacol. 2023;14:1162889. doi: 10.3389/fphar.2023.1162889

**Table 4 ijms-26-08386-t004:** Strategies for Optimizing the Use of Rodent Models of Lung Disease.

Optimization Strategy	Description	Potential Benefits	Challenges
Combining toxins or insults	Use multiple agents (e.g., elastase + LPS, or bleomycin + cigarette smoke) to emulate complex pathologies.	Mimics multifactorial human disease; induces robust, heterogeneous injury.	Potential for confounding interactions.
Altering temporal sequence of insults	Adjust timing between exposures (e.g., preconditioning vs. simultaneous vs. sequential insults).	Reveals priming effects, injury progression, and recovery mechanisms.	Requires precise scheduling; may complicate interpretation of outcomes.
Delaying therapeutic intervention	Introduce treatment after disease is well-established (e.g., 7–21 days post-injury).	Reflects realistic clinical scenarios; allows evaluation of regeneration mechanisms.	Requires longer studies; potential for irreversible damage before treatment.
Comparative model analysis	Test multiple models (e.g., elastase vs. bleomycin; transgenic vs. toxin exposure) for same disease context.	Distills mechanistic relevance; improves translational accuracy.	Strain, time course, or injury severity may confound comparisons.
Stratifying by injury outcome	Focus on specific outcomes (e.g., fibrosis vs. emphysema), or pathological features.	Enhances precision and rigor; allows targeted mechanistic insights.	Requires extensive phenotyping; may exclude intermediate/combined outcomes.
Integrating genetic manipulation	Combine with transgenic or knockout models.	Clarifies gene-environment interactions; enables mechanistic dissection.	Genetic background may influence response; complex breeding required.

## Data Availability

Not applicable.

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
