# Peer review of "Rodent Models of Lung Disease: A Road Map for Translational Research"

_ijms, 2025, doi:10.3390/ijms26178386_

Round 1
Reviewer 1 Report
Comments and Suggestions for Authors
In my opinion, the topic stated in this manuscript, animal models of lung diseases, may be very interesting and current, and a comprehensive overview focused on one of the many key topics in this field of research would be very useful for researchers facing a related preclinical project, supporting the appropriate choice of the most suitable animal model that best fits the specific scientific question, as well as the most appropriate methodology and experimental design. It is worth considering the advantages and limitations of the animal models mentioned, as well as the challenges related to comparative medicine and ethical aspects, as well as current refinements aimed at safeguarding both animal welfare and the quality of science. Overall, the English language of the manuscript is clear and some of the most commonly used methods for generating animal models of lung diseases have been mentioned.
However, in my opinion there are a series of major revision concerns that should be addressed before reconsidering the manuscript for publication.
First, it is necessary to focus the topic more clearly, both in the title and in the text of the manuscript: in the proposed manuscript only some laboratory (induced) rodent lung disease models (hamster, mouse) are considered. Lung disease models, both spontaneous and induced (by chemical or genetic manipulation), exist in several mammalian species. Discussion of lung disease models other than chemically induced models in laboratory rodents may be appropriate to defer to some relevant articles, clearly stating their objectives. Even in a comprehensive narrative review, it is necessary to provide a description of the search strategy and mesh terms, clearly defining the animal models described (species, strains), the methodology to generate them (spontaneous, induced, genetically modified, etc.), their advantages and limitations, and the scientific and animal welfare issues. In my opinion, it is appropriate to describe one's experience in the manuscript (but not in the abstract and introduction!) if the research group in question has specific expertise in this specific field of research. However, approximately 18% of the bibliographic references are represented by publications of the author's research group and/or collaborators, and these references are those most extensively detailed in the text of the manuscript. I would suggest balancing both the number and the description of the articles, taking into account the most relevant authors in relation to the different rodent models treated. Furthermore, I would suggest to better clarify the objectives of the review in the introduction and to organize the division of the paragraphs more rationally (e.g. 2. models of acute lung injury (brief introduction to the type, mode of generation, etc.); 2.1 LPS; 2.2 cigarette smoking; 3. models of emphysema; 3.1 models of elastase-induced emphysema; 3.2 models of cigarette smoking-induced emphysema; etc.). Finally, I would suggest specifically addressing current ethical issues and refinements achieved in the indicated animal models, to ensure improvements in both animal welfare and the quality of science.
In light of these premises, I would suggest to reconsider submission after major revisions and provide a list of recommendations aiming to improve the manuscript:
- Title: The Strategic Use of Animal Models in Translational Research
I would first suggest making the title less generic, focusing on “rodent models of lung disease” as mentioned above, to capture the attention of researchers interested in the specific topic.
Furthermore, in my opinion "The strategic use..." sounds misleading and at least "anthropocentric" in a "one health, one medicine" perspective: would the author reconsider these terms, for example "The current role of rodent models in pulmonary research"; The complementary role of..." or similar?
For the same reasons, could you reconsider “limiting” the models indicated to laboratory rodents in the abstract and manuscript? - Abstract: In many cases, the morphological changes associated with a particular model may resemble those seen in human disease, but the corresponding biochemical events' nature and timeframe may differ. Nevertheless, gaining insight into human disease mechanisms may be possible by employing experimental approaches that minimize the problems associated with extrapolating data from animal studies. Such strategies may include using more than one model of a particular disease, employing different routes of administration of the injurious agent, using a variety of animal strains or species, or focusing on biochemical mechanisms common to both the animal model and its human counterpart. This last strategy is particularly useful in testing potential therapeutic agents or developing biomarkers for human lung disease. In my opinion, this initial part of the abstract correctly highlights critical points like specie-specific similarities and differences across animal models and humans, which determine their potential role in providing insights to clinical setting. These comparative aspects include not only the anatomical, physiological, histological or molecular characteristics of the lesions, but also the pharmacokinetics and pharmacodynamics of the tested compounds. An effective research strategy may include the complementary use of in vitro and in vivo models, the latter chosen rationally based on the advantages and limitations of each one from a comparative medicine perspective, as well as in relation to the specific research question addressed. I think that these concepts should be clearer described, avoiding misleading sentences: for example, author could rephrase “using more than one model of a particular disease, different routes of administration of the injurious agent, using a variety of animal strains or species,” such as “the integrate, rational choice of the proper animal model to address a specific research question, as well as a critical definition of experimental methods and design”.
- Abstract: For example, we have utilized animal models of emphysema that specifically involve signicant elastic fiber injury because the breakdown of these fibers is a critical component of the human disease. As a result, we developed a therapeutic approach to mitigating elastic ber injury that may slow the progression of human pulmonary emphysema. The same models were also useful in testing a new biomarker for the disease that may be a be fiber indicator of the efficacy of novel therapeutic agents. Thus, the appropriate use of animal models is critically important in translating experimental findings to the clinical setting. Of course, I think you can say that you will also describe your personal experiences, but even if it is a narrative review, you should clearly describe the purpose (animal model? disease? mechanism? therapies?), the literature search strategy and criteria (database, time frame, keywords and mesh terms, etc.), and the general results of the data extracted from the selected articles.
- Introduction: Animal models facilitate...could author consider an alternative term like contribute? Furthermore, I would suggest that you do not start with examples from your own works at the beginning, but introduce the species, disease patterns, ethical concerns you intend to describe, as well as general similarities and limitations in comparative research.
- Paragraphs 2 -5: As mentioned above, I suggest you organize your paragraphs more clearly based on the disease model you are dealing with, the method of generating these models you describe, the species in which they are reproduced, the main comparative aspects with humans, and the main advantages and limitations. I suggest you to make a brief introduction on each of the main model generation mechanisms mentioned before describing in detail the results of your works and those of other relevant authors.
- Paragraph 6: I would suggest avoiding a separate paragraph on this topic, while the general advantages and limitations of chemically induced lung disease models compared to genetically modified ones could be discussed in the introduction, specifying that you will talk about the former and that detailed information on GEMMs could be found in other relevant articles [REF].
- Paragraph 7: I would suggest discussing in general terms the main evidence from all relevant references cited for the "review" (I would suggest avoiding the phrase "The current article shows" as this is a narrative review, not a research article).
- Figures: in my opinion, figures should be more informative and legends should be more detailed. For example, Fig1 should be informative of comparative similarities of LPS (mouse?) model and acute lung inflammation in humans, as well as of the species, the technique used for the preparation of the sample showed, and use arrows or other ways to indicate cells and morphological structures described in the image. Furthermore, Fig. 2 - 5 and 7, 9-11 could be more useful if associated with indication of pathways involved, main references, etc. Finally, I would suggest to avoid figures like 6, 8, 12 showing results of your own researches.
Author Response
Response to Reviewers
Note: The responses are listed in the same order as the reviewers’ comments. Revisions to the manuscript are highlighted in gray.
Reviewer #1:
- The title of the manuscript has been revised to more specifically represent the content of the review.
- The Introduction has been rewritten to emphasize the objectives of the review and provide an overview of the advantages, limitations, and the translational role of rodent animal models (pp 1-2). This discussion is now supplemented with the addition of a table (4).
- The discussion of animal models is now exclusively focused on rodents, however a table (3) has been added that lists some of the advantages and disadvantages of specific genetic models.
- The abstract has been modified to reflect the revisions to the manuscript (p 1).
- The rationale for the specific use of the animal models included in the manuscript is described in greater detail (multiple pages), along with a discussion of ethical issues (pp 2, 16).
- The section describing genetic models of lung disease has been removed from the manuscript and replaced with a more concise discussion of these models in the Introduction (p 2).
- The term, “current paper,” has been deleted from the manuscript, and replaced by “review.”
- The figures have been revised, including the captions, to more specifically describe the associated animal models and disease mechanisms related to the figures.
- Figures 8 and 12 have been deleted and replaced by others representing general disease mechanisms rather then specific experimental data. Figure 6 (fig 5 in revised manuscript) was retained based on comments by reviewer #2.
Reviewer #2:
- The title of the manuscript has been revised to better reflect the content of the review.
- The Abstract have been modified to more specially describe the advantages and disadvantages of animal models and their role in translational research.
- A table (3) has been added to the revised manuscript that presents the advantages and disadvantages of in vivo and in vitro lung disease models.
- A section discussing single-cell RNA sequencing has been included in the revised manuscript (pp 15-16).
- The new table 3 provides the comparative strengths and weaknesses of various categories of animal models. The table specifically focuses on rodent models in accordance with comments by other reviewers.
- A discussion of the failures of certain animal models to accurately represent human disease mechanisms, thereby limiting their usefulness in translational research, is included in the revised manuscript (multiple pages). Additional information regarding the limitations of these models is included in table 3.
- The relevant captions have been revised to include specific experimental conditions associated with the results expressed in the figure.
Reviewer #3:
- The last sentence of the Abstract has been revised to more closely reflect the content of the manuscript.
- The last paragraph of the Introduction now includes a more definitive statement regarding the ways in which the animal research in the review “focuses on how to transform the role of these models from their traditional role in describing morphological and biochemical changes to a more dynamic platform that can identify critical mechanisms of pulmonary injury, interactions of toxic agents, and potential treatment targets.” (p 2). In the subsequent sections of the manuscript, the discussion of specific studies is framed in terms of this perspective (multiple sections).
- The limitations of the LPS model is more fully discussed (p 3, last paragraph). Similarly, accurately interpretation of the results of therapeutic interventions in the BLM model may be impaired by the spontaneous regression of the disease over time (p 13, paragraph 4).
- The disparities between the reviewed animal models and their human counterparts are more fully discussed in the revised manuscript, including the addition of a table (4) listing some of the main differences.
- The term “inflammatory index” is now better defined (p 4).
- Potential mechanisms responsible for the synergy between cigarette smoke and LPS are more fully discussed in the revised manuscript (p 4-5). Similarly, mechanisms are proposed for the synergistic interaction between elastase and LPS (p 9, last paragraph; figure 7).
- The acronym, “ERA” is introduced with the first appearance of the term, “endothelin receptor antagonist” (p 5).
- The term, “gatekeeper,” has been replaced with a more physiological descriptor for the effect of endothelin (p 5, and figure 3 caption).
- The revised manuscript contains a more complete description of the mechanisms involved in the inhibition of endothelin by HJP272 and phosphoramidon (p 6).
- The clinical applicability of the free to bound DID ratio is discussed in the revised manuscript (p 8).
- The rationale for selecting the molecular weight and dose of HA is provided in the revised manuscript (pp 11, 12).
- The concept of “emergence” is discussed more fully (pp 14, 15) and accompanied by a table (2) presenting the multiscale features of this process.
- More recent citations regarding the use of ERAs to treat human pulmonary fibrosis have been added to the manuscript.
- The section on chemical vs genetic models has been removed at the request of another reviewer. However, a discussion of the role of genetic models has been added to the Introduction (p 2) and specific genetic models are included in table 1 and 3.
- The reference to synergistic interactions has been replaced by “synergy” (p 19).
- The spelling errors have been corrected.
- Scale bars could not be inserted in the photomicrographs due to a current lack of information regarding the specific characteristics of the microscope used to photograph the images. However, the original magnification has been added to the captions.
- Several figures or their associated captions have been revised to provide a better context for their inclusion in the manuscript.
- The relevant captions now include the term, “photomicrographs” (figures 1, 8, 9).
- All figures have been adjusted for maximum resolution.
Reviewer #4:
- The section discussing genetic models has been removed at the request of another reviewer, but is replaced by table 1, which list the characteristics of multiple rodent knockout models.
- The revised manuscript discusses the use of specific rodent species for certain models of lung injury (pp 18-19).
- The use of black and white photomicrographs was based on the availability of appropriate images.
- The revised manuscript discusses the role of endothelin and its cellular sources (p 6).
- The revised Introduction and other sections of the manuscript more closely relate the role of inflammation in the pathogenesis of lung disease (multiple pages).
- The mechanism responsible for the absorption of energy by elastin is discussed in greater detail (p 8).
- The acronym “BALF” is now associated with the first appearance of the full terminology in the text.
- The details of cigarette smoke exposure is described in further detail (p 12).
- The use of the acronym, ERA, has been modified as suggested.
- The original figure 8 was removed from the revised manuscript as requested by another reviewer.

Reviewer 2 Report
Comments and Suggestions for Authors
In the present review article "The Strategic Use of Animal Models in Translational Research", Dr. Cantor provides a comprehensive and well-organized overview of various animal models used in studying lung disease. However, the manuscript risks reinforcing a purely pro-animal-model view without sufficient critical balance or ethical framing. To increase the impact and clarity of this review, several important edits are recommended, including the 3Rs (Replacement, Reduction, Refinement) and alternative models, as well as relevance for human studies, including clinical translation successes and failures.
- As the title broadly suggests "a strategic use in translational research", the current lung-specific focus might seem narrower than implied. Please consider adjusting the title to specify lung disease models, or adding a short section or paragraph acknowledging other systems.
- The first and the last sentences of the abstract are too vague and should be rephrased. The first sentence also sounds a bit misleading as the relevance of animal models to human diseases is not uncertain, rather it is well designed and relevant.
-
The review must acknowledge and discuss alternative approaches that can reduce or replace animal use in lung disease research. Please consider adding a clear subsection covering alternatives such as 3D human lung organoids, lungs on the chip, human ex vivo lung tissue slices, computational modeling in silico methods.
- The current manuscript omits any discussion of single-cell RNA sequencing (scRNA-seq), which has become a critical and widely adopted tool in modern translational lung research. I strongly recommend adding a subsection on this topic to strengthen the review’s currency and relevance. A cross-species integration of scRNA-seq data is now routinely used to compare mouse model responses to human disease, identifying both conserved and divergent pathways. The limitations of this approach are also present and should be included (difference in gene expression between homologous cell types, inconsistencies in matching developmental stages or injury severity between species...). Including the scRNA-seq atlases of murine and human fibrotic lungs would improve the review article significantly.
- The manuscript would benefit greatly from a clear, structured table summarizing different animal models and species (e.g., mice, hamsters, rats, dogs, primates) and target lung conditions for each animal model. Include pros and cons for each model (cost, effect, similarity to human system...).
- The section addressing limitations is too descriptive. Please add a clear statements to address the points where have animal models failed in translation and why.
- All figure legends should include species used and precise experimental conditions (dose, timing, groups, number of animals per group).
-
Author Response

(The authors gave the same response as above.)

Reviewer 3 Report
Comments and Suggestions for Authors
General Assessment
The manuscript by Jerome Cantor provides a timely and comprehensive overview of the strategic use of animal models in lung disease research, highlighting their role in translational studies for therapeutic development and biomarker discovery. The author effectively discusses major models, including LPS-induced ALI, elastase-induced emphysema, cigarette smoke-induced emphysema, and bleomycin-induced fibrosis, while highlighting interventions such as aerosolized hyaluronan (HA) and endothelin receptor antagonists (e.g., HJP272). Additionally, the paper proposes the free-to-bound desmosine/isodesmosine (DID) ratio as a biomarker to improve translational relevance and addresses therapies (HA, ERAs) for COPD and pulmonary fibrosis. However, some sections require clarification, expansion, or deeper mechanistic insights (e.g., molecular pathways in synergistic models) due to incomplete methodological details and unclear novelty, which would enhance the manuscript’s overall quality and impact.
Comments:
The abstract effectively summarizes the manuscript. However, the last sentence "Thus, the appropriate use of animal models is critically important in translating experimental findings to the clinical setting" (lines 24-25) could be rephrased to emphasize the strategic and thoughtful application discussed in the paper, rather than just "appropriate use," to align more closely with the manuscript's title.
While the review is well-structured, it largely summarizes established models without offering significant new insights or critical analysis and does not sufficiently differentiate its contributions from prior work (e.g., references 1–3, 8–9, 41–44). The introduction does not clearly articulate a specific hypothesis or research question, reducing focus. Clearly articulate how the study’s findings advance beyond existing literature.
Balancing Strengths and Limitations: The manuscript generally does a good job of presenting both the strengths and limitations of each model. However, in some sections, the discussion of limitations feels somewhat brief compared to the detailed description of the model's utility. For instance, in the LPS model, while it's mentioned that it "may not fully encompass chronic lung injury or repair processes" (line 74), expanding on why this is the case and its implications for translational research would be beneficial. Similarly, for the bleomycin model, while the "wound-healing phenomenon" and "regression over time" are noted, a deeper dive into how these aspects limit its direct human relevance for chronic fibrosis would be valuable.
Refine the introduction and conclusions to focus on specific contributions and limitations. Expand on the caveats of extrapolating data from animal models in the discussion sections for each model, particularly regarding chronicity and multifactorial aspects of human disease.
Define "inflammatory index" (line 101) and clarify how it is quantified.
The synergistic effects of cigarette smoke and LPS (Section 2.2) and LPS/elastase (Section 3.3) are compelling but are described phenomenologically and require more detailed explanation. Elaborate on shared inflammatory pathways (e.g., oxidative stress, inflammasome activation) to strengthen mechanistic insights.
Lines 123-124: “Our studies showed that pretreatment with a novel endothelin receptor antagonist (HJP272)” → “Our studies showed that pretreatment with HJP272, a novel endothelin receptor antagonist (ERA)”. Introduce the term ERA as it is used for the first time.
Replace "gatekeeper for neutrophils" (line 130) with a more precise descriptor (e.g., "chemoattractant", “adherence or attachment mediator”).
The role of HJP272 and phosphoramidon in ALI and fibrosis is promising but lacks mechanistic depth (e.g., specific endothelin receptors targeted). Elaborate on the molecular mechanisms of HJP272 and phosphoramidon, specifying targeted receptors and pathways.
The negative correlation between the free-to-bound DID ratio, proposed as a biomarker, and lung surface area in elastase-induced emphysema (Section 3.2) is a robust finding, supported by statistical analysis. However, its clinical applicability remains speculative. Include comparative data from human studies (if available) or discuss challenges in validation (e.g., sensitivity in early-stage disease).
The use of aerosolized HA to reduce airspace enlargement in both elastase and cigarette smoke models (Section 4.2) is innovative, and HA’s protective effects against elastic fiber injury in emphysema models are promising and validated across acute (elastase) and chronic (cigarette smoke) models, with translational relevance shown in a clinical trial for COPD patients with alpha-1 antitrypsin deficiency. However, the rationale for its aerosolized use (dosing with a 0.1% solution and a 150 kDa molecular weight) lacks justification. Compare with other mucoprotective agents if possible.
The bleomycin model is described as a wound-healing phenomenon (Section 5.1) but lacks quantitative data to support claims about fibrosis regression. Include quantitative data for the bleomycin model to support the dynamics of fibrosis.
ERAs’ failure in clinical fibrosis (Section 5.2) contrasts with preclinical success. Discuss potential reasons (e.g., timing, pharmacokinetics) to reconcile discrepancies. The concept of "emergence" in fibrosis is intriguing but underdeveloped. Clarify how percolation theory applies mechanistically to fibrosis progression. Citations are outdated (e.g., references to "recent" clinical trials should be post-2020).
The "Chemical vs. Genetic Models" section (Section 6) feels tangential. Integrate this comparison earlier in the Introduction or expand to discuss hybrid approaches (e.g., smoke exposure in CFTR knockouts).
The discussion on limitations (Section 7) is general and superficial; it could be more specific about model-specific shortcomings, and a deeper critique of model discrepancies (e.g., interspecies differences, dose-response variations) would enhance translational relevance.
Line 396: "synergistic interaction of this agent" → "synergy between these agents."
The manuscript contains minor typographical errors (e.g., "interieukin-1β" in reference 4). Correct typographical errors and ensure consistent terminology.
Figure 1 (LPS model): Add scale bars and quantify neutrophil influx (e.g., cells/mm²).
Figures 2, 3, and 4 are illustrative but do not provide more detailed information than the text, hindering their utility.
Figures 9, 10, and 12: Add scale bars
Figures 9 and 10: Begin their description with what is shown, “Photomicrographs of”
Ensuring all figures meet the journal's high-resolution requirements would improve visual quality.
Ensure consistent referencing throughout the manuscript. While most references are in numerical order, double-check for any inconsistencies and completeness (reference 30 is incomplete).
Author Response

(The authors gave the same response as above.)

Reviewer 4 Report
Comments and Suggestions for Authors
The manuscript entitled “The Strategic Use of Animal Models in Translational Research” is well written in clear understandable English and focus the attention of the reader on different models (first of all chemically-induced) of lungs disorders. In general, the manuscript is well organized and authors covers multiple questions. However, I have multiple questions which can be divided in two groups: Major and Minor.
Major
- The Chapter #6 should be enlarged. There are many types of genetically-related models of respiratory system disorders. For example, Ace2 knockouts; IL11RA1 knockout; CAF1-knockout, and many others. You also should discuss in more details application of genetically modified animals for studying respiratory diseases. It also will be good if you will make some table in this chapter which will briefly descripts existing knockouts and diseases which they reproduce.
- Also, you should enlarge chapter Limitations. As I know, some respiratory diseases cannot be reproduced on typical laboratory animals such as mice, rats and hamster due to that their respiratory system is significantly shorter in comparison to human. Therefore, for example, Cystic fibrosis cannot be reproduced in small animals. It will be interesting to read in your manuscript these kinds of limitations: when and why some laboratory animal cannot be used for testing because it is bad human model. Also, you describe that in some experiments were involved hamsters but in other experiments mice were used. What is the reason of this difference? As I know there are differences in modulating COVID in mice and hamsters. You also may make some accent in this field.
Minor
- Figures 1 and 9. Why this Figures are black and white? Which method was used for obtaining these Figures? Histology? If yes, why they are B and W? Which staining methods were used?
- Figure 3. Is endothelin released from the alveoli or from the blood vessel? Does the endothelin-receptor complex penetrate the blood vessel? Does this interfere with blood flow?
- In the Introduction section you should more focus attention of the reader why it is required to reduce inflammation on lungs. When human is ill inflammation is a normal process.
- Lines 148-149. Explain how they absorb energy. Do they bind ATP molecules?
- Line 126. There is the first mention of acronym BALF. Provide full term here.
- Line 217. Describe in more details the experiment. Animals received smoke for 2 hours permanently? Or, for example, 10 minutes each 2 hours? What was the intensity of the smoke?
- Line 306. Acronym ERAs? Deabbreviate.
- Lines 261-268. Such treatment resulted improving the well-being?
- Figure 8. Indicate statistical differences by asterisks.
Author Response

(The authors gave the same response as above.)

Round 2
Reviewer 1 Report
Comments and Suggestions for Authors
line 45 (typos): 6)
line 50: it would be further improve the manuscript if author could insert references about models described in table 1, since he introduce them, but thereafter specify that her review will focus on rodent models of chemically induced lung diseases.
line 65-68 (typos): repeated words/concept "rather than larger mammals" "than using larger mammals or humans"; furthermore, I would suggest to rephrase , e.g. " Using rodent models, rather than larger mammals, may be more ethically acceptable and practically feasible for broad exploration of potential therapies and mechanisms in shorter timescales and at lower husbandry costs". Please, consider similar suggestion for table 2.
line 125, section 2.2.: please format your text as above (justified).
line 182-188, section 2.3.: since this review has a single author, I would suggest avoiding the opening phrase "We..", "our study".
line 324, section 4.1.: please format your text as above (justified).
line 418, section 5.1.: please format your text as above (justified).
line 499-525: there are two section 6, please correct
line 499: In my opinion, the inclusion of a separate section on a specific analysis technique should be more clearly justified and improved than in its current form. I would suggest improving the title first, focusing on rodent models of lung disease. Furthermore, I suggest that the author better explain the reasons why he specifically report only this single technique, excluding other equally useful for in vivo or ex vivo characterization of rodent models of lung diseases e.g. in vivo uCT or optical imaging. For example, you might contextualize this specific choice in relation to its frequency of use in the references cited above, or to evidence its role as a gold standard in the field of research addressed.
line 525, section 6 (bis) "Animal models alternatives": As stated previously in section 6 on single-cell RNA sequencing, in my opinion this section does not seem clearly connected to the previous general sections. Furthermore, if the title is "Animal models alternatives", the Table 3 inserted in this paragraph, containing description of both animal models and in vitro alternatives, is not clear. To improve clarity of tthe manuscript, I would suggest that authors carefully review the last 3 sections (6 "single-cell RNA sequencing", 6 "Animal models alternatives" and 7 "Strategies for Addressing the Limitations of Animal Models") to elaborate a more clear, well contextualized sinthesis (a unique section could also work well) highlighting advantages and limitations of the animal models discussed and the potential usefulness of methodologies such as single-cell RNA sequencing or alternatives to maximize information from these animal models.
-line 624: the term "The current paper" has not been replaced by review at this part. Please, carefully check the manuscript.
Author Response
Response to Reviewer Comments
Note: Revisions to the manuscript are highlighted in gray. Responses are listed in the same order as the reviewer comments.
Reviewer #1
- The typographical error on line 45 has been corrected.
- The revised manuscript includes a paragraph discussing the specific models listed in table 1, and references have been inserted in the table (lines 40-52). The text also includes a statement that the review focuses on chemically induced rodent models (lines 55-61).
- The sentence regarding the ethical and economic advantages of rodent models has been revised as suggested by the reviewer (lines 70-74).
- Figure 2 caption text has been reformatted to full justification.
- The terms “we” and “our” as a reference to the author have been removed from the manuscript.
- The additional text justifications have been included in the revised manuscript.
- The sections have been renumbered to remove the error.
- The title for section 6 has been revised, and now includes the rationale for using single-cell RNA sequencing (lines 506-526).
- Sections 6 and 7 have been revised to better contextualize their relationship to the other components of the manuscript (particularly with regard to single-cell RNA sequencing).
- The term “current paper” has been replaced by “current review” (lines 569, 627, and 648).
Reviewer 4 Report
Comments and Suggestions for Authors
- Tables 1 and 3 require references.
- In vitro should be italic across all the text
Author Response
Response to Reviewer Comments
Note: Revisions to the manuscript are highlighted in gray. Responses are listed in the same order as the reviewer comments.
Reviewer #2
- References have now been added to tables 1 and 3.
- The term “in vitro” has been italicized.